# Autoantibody discovery across monogenic, acquired, and COVID-19-associated autoimmunity with scalable PhIP-seq

Sara E Vazquez[1,2,3†], Sabrina A Mann[1,4†], Aaron Bodansky[5†], Andrew F Kung[1], Zoe Quandt[2,6], Elise MN Ferré[7], Nils Landegren[8,9], Daniel Eriksson[10,11], Paul Bastard[12,13,14,15], Shen-Ying Zhang[12,13,14], Jamin Liu[1,16], Anthea Mitchell[1,4], Irina Proekt[2], David Yu[2], Caleigh Mandel-Brehm[1], Chung-Yu Wang[1,4], Brenda Miao[1], Gavin Sowa[3], Kelsey Zorn[1], Alice Y Chan[17], Veronica M Tagi[18], Chisato Shimizu[19], Adriana Tremoulet[19], Kara Lynch[20,21], Michael R Wilson[22], Olle Kämpe[8,23,24], Kerry Dobbs[25], Ottavia M Delmonte[25], Rosa Bacchetta[18], Luigi D Notarangelo[25], Jane C Burns[19], Jean-Laurent Casanova[12,13,14,15,26], Michail S Lionakis[7], Troy R Torgerson[27,28§], Mark S Anderson[2*‡], Joseph L DeRisi[1,4*‡]

*For correspondence:
Mark.Anderson@ucsf.edu (MSA);
joe@derisilab.ucsf.edu (JLDeR)

†These authors contributed
equally to this work
‡These authors also contributed
equally to this work

Present address: §Allen Institute
for Immunology, Seattle, United
States

Competing interest: The authors
declare that no competing
interests exist.

Reviewing Editor: Antony
Rosen, Johns Hopkins University
School of Medicine, United
States

[1]Department of Biochemistry and Biophysics, University of California, San Francisco, San Francisco, United States; [2]Diabetes Center, University of California, San Francisco, San Francisco, United States; [3]School of Medicine, University of California, San Francisco, San Francisco, United States; [4]Chan Zuckerberg Biohub, San Francisco, United States; [5]Department of Pediatric Critical Care Medicine, University of California, San Francisco, San Francisco, United States; [6]Department of Medicine, University of California, San Francisco, San Francisco, United States; [7]Fungal Pathogenesis Unit, Laboratory of Clinical Immunology & Microbiology, National Institute of Allergy and Infectious Diseases, National Institutes of Health, Bethesda, United States; [8]Department of Medicine, Karolinska University Hospital, Karolinska Institute, Stockholm, Sweden; [9]Science for life Laboratory, Department of Medical Sciences, Uppsala University, Uppsala, Sweden; [10]Department of Medical Biochemistry and Microbiology, Uppsala University, Uppsala, Sweden; [11]Centre for Molecular Medicine, Department of Medicine, Karolinska Institutet, Stockholm, Sweden; [12]St. Giles Laboratory of Human Genetics of Infectious Diseases, Rockefeller University, New York, United States; [13]Laboratory of Human Genetics of Infectious Diseases, Necker Branch, INSERM U1163, Necker Hospital for Sick Children, Paris, France; [14]Imagine Institute, University of Paris, Paris, France; [15]Department of Pediatrics, Necker Hospital for Sick Children, Paris, France; [16]Berkeley-University of California, San Francisco Graduate Program in Bioengineering, University of California, San Francisco, San Francisco, United States; [17]Department of Pediatrics, Division of Pediatric Allergy, Immunology, Bone and Marrow Transplantation, Division of Pediatric Rheumatology, University of California, San Francisco, San Francisco, United States; [18]Division of Stem Cell Transplantation and Regenerative Medicine, Stanford University School of Medicine, Stanford, United States; [19]Kawasaki Disease Research Center, Rady Children's Hospital and Department of Pediatrics, University of California, San Diego, La Jolla, United States; [20]Department of Laboratory Medicine, University of California, San Francisco, San Francisco, United States;

[21]Zuckerberg San Francisco General, San Francisco, United States; [22]Weill Institute for Neurosciences, University of California, San Francisco, San Francisco, United States; [23]Department of Clinical Science and KG Jebsen Center for Autoimmune Disorders, University of Bergen, Bergen, Norway; [24]Center of Molecular Medicine, and Department of Endocrinology, Metabolism and Diabetes, Karolinska University Hospital, Stockholm, Sweden; [25]Laboratory of Clinical Immunology and Microbiology, National Institute of Allergy and Infectious Diseases, National Institutes of Health, Bethesda, United States; [26]Howard Hughes Medical Institute, New York, United States; [27]Seattle Children's Research Institute, Seattle, United States; [28]Department of Pediatrics, University of Washington, Seattle, United States

**Abstract** Phage immunoprecipitation sequencing (PhIP-seq) allows for unbiased, proteome-wide autoantibody discovery across a variety of disease settings, with identification of disease-specific autoantigens providing new insight into previously poorly understood forms of immune dysregulation. Despite several successful implementations of PhIP-seq for autoantigen discovery, including our previous work (Vazquez et al., 2020), current protocols are inherently difficult to scale to accommodate large cohorts of cases and importantly, healthy controls. Here, we develop and validate a high throughput extension of PhIP-seq in various etiologies of autoimmune and inflammatory diseases, including APS1, IPEX, RAG1/2 deficiency, Kawasaki disease (KD), multisystem inflammatory syndrome in children (MIS-C), and finally, mild and severe forms of COVID-19. We demonstrate that these scaled datasets enable machine-learning approaches that result in robust prediction of disease status, as well as the ability to detect both known and novel autoantigens, such as prodynorphin (PDYN) in APS1 patients, and intestinally expressed proteins BEST4 and BTNL8 in IPEX patients. Remarkably, BEST4 antibodies were also found in two patients with RAG1/2 deficiency, one of whom had very early onset IBD. Scaled PhIP-seq examination of both MIS-C and KD demonstrated rare, overlapping antigens, including CGNL1, as well as several strongly enriched putative pneumonia-associated antigens in severe COVID-19, including the endosomal protein EEA1. Together, scaled PhIP-seq provides a valuable tool for broadly assessing both rare and common autoantigen overlap between autoimmune diseases of varying origins and etiologies.

## Editor's evaluation

This is an important paper that is methodologically compelling. The work presents a series of enhancements to the PhIP-seq method of autoantibody discovery, improving scaling to larger cohorts and control populations, and increasing the ability to discover disease-specific immune responses. The approach is used to discover a novel, frequent autoantibody response (BTNL8) in IPEX patients, and will be an accessible approach to investigate the presence and specificity of auto-antibodies in diseases where these have been difficult to define.

## Introduction

Autoantibodies provide critical insight into autoimmunity by informing specific protein targets of an aberrant immune response and serving as predictors of disease. Previously, disease-associated auto-antigens have been discovered through candidate-based approaches or by screening tissue specific expression libraries. By contrast, the development of proteome-wide screening approaches has enabled the systematic and unbiased discovery of autoantigens.

Two complementary approaches for proteome-wide autoantibody discovery include printed protein arrays and phage-immunoprecipitation sequencing (PhIP-seq) (*Jeong et al., 2012*; *Larman et al., 2011*; *Sharon and Snyder, 2014*; *Zhu et al., 2001*). While powerful, printed protein arrays can be cost- and volume-prohibitive and are not flexible to adapting or generating new antigen libraries. On the other hand, PhIP-seq, originally developed by *Larman et al., 2011*, uses the economy of scale of arrayed oligonucleotide synthesis to enable very large libraries at comparatively low cost. However,

phage-based techniques have remained hindered by labor-intensive protocols that prevent broader accessibility and scaling.

Recently, we and others have adapted and applied PhIP-seq to detect novel, disease-associated autoantibodies targeting autoantigens across a variety of autoimmune contexts (*Larman et al., 2011*; *Mandel-Brehm et al., 2019*; *O'Donovan et al., 2020*; *Vazquez et al., 2020*). However, both technical and biological limitations exist. From a technical standpoint, PhIP-seq libraries express programmed sets of linear peptides, and thus this technique is inherently less sensitive to detect reactivity to conformational antigens, such as Type I interferons (IFNs) and GAD65 (*Vazquez et al., 2020*; *Wang et al., 2021*). Nonetheless, the technique allows hundreds of thousands to millions of discrete peptide sequences to be represented in a deterministic manner, including the multiplicity of protein isoforms and variants that are known to exist in vivo. In this sense, PhIP-seq is complementary to mass spectrometry and other techniques that leverage fully native proteins. Given that many forms of autoimmunity exhibit significant phenotypic heterogeneity, the true number of patients with shared disease-associated autoantibodies may be low (*Ferre et al., 2016*). Therefore, the screening of large cohorts is an essential step for identifying shared antigens and would benefit from a scaled PhIP-seq approach for high throughput testing.

Beyond the benefits of detecting low-frequency or low-sensitivity antigens, a scaled approach to PhIP-seq would also facilitate increased size of healthy control cohorts. Recently, PhIP-seq has been deployed to explore emerging forms of autoimmune and inflammatory disease, including COVID-19-associated multisystem inflammatory syndrome in children (MIS-C) (*Gruber et al., 2020*). However, these studies suffer from a relative paucity of control samples, resulting in low confidence in the disease specificity of the suggested autoantigens. Questions of disease specificity, rare antigens, and antigen overlap can be addressed in larger, scaled experiments.

Here, we develop a high-throughput PhIP-seq protocol with markedly increased accessibility (not requiring robotics) and scale (enabling 600–800 samples to be run in parallel), with minimal

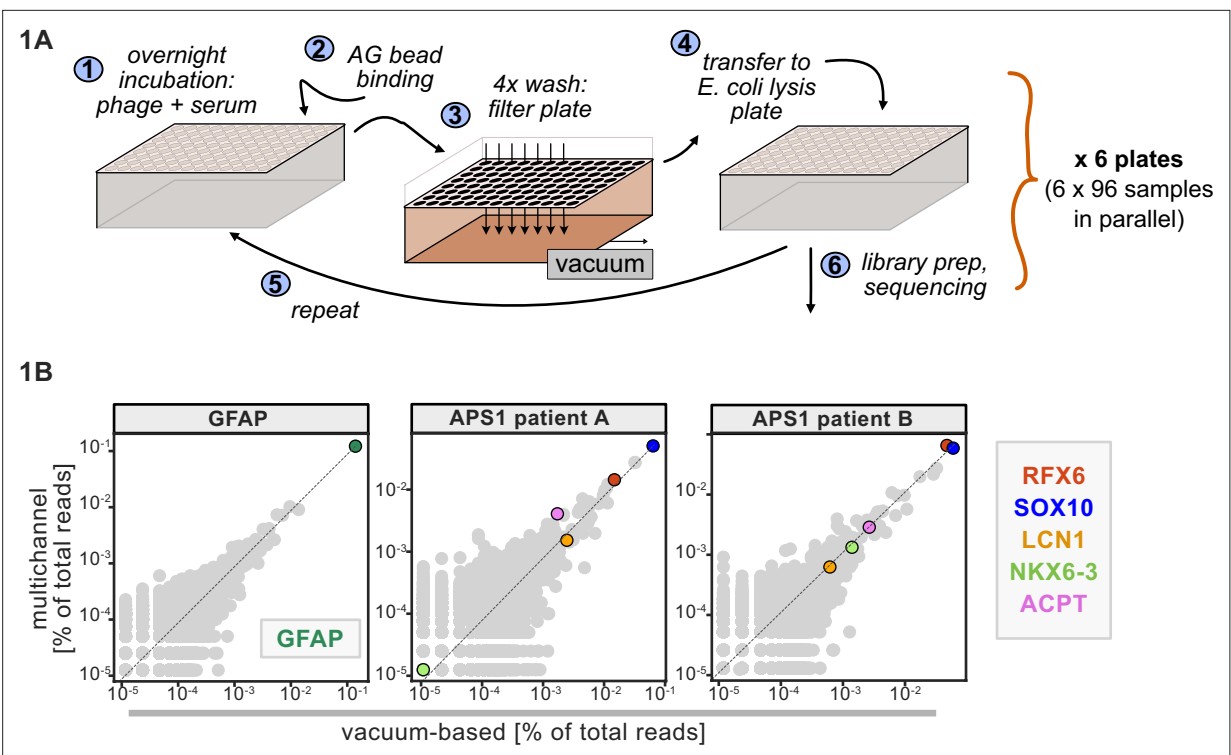

**Figure 1.** Advantages of and considerations motivating scaled phage-immunoprecipitation sequencing (PhIP-seq). (**A**) Schematic of vacuum-based scaled PhIP-seq protocol, allowing for parallelized batches of 600–800 samples. (**B**) Comparison of moderate-throughput multichannel protocol data to high-throughput vacuum-based protocol data, with axes showing normalized read percentages. Controls include a commercial polyclonal anti-GFAP antibody (left), APS1 patient A with known and validated autoantibodies RFX6, SOX10, ACPT, and LCN1 (center), and APS1 patient B with the same known and validated autoantibodies as well as NKX6-3.

plate-to-plate variability and low contamination potential, and without sacrificing data quality. We demonstrate the utility of this protocol in the context of an expanded, multi-cohort study, including APS1, patients with immune dysregulation, polyendocrinopathy, X-linked (IPEX), RAG1/2 deficiency with autoimmune features, a KD patient cohort, and emerging COVID-19 patient phenotypes with possible autoimmune underpinnings. In the future, scaled PhIP-seq cohort studies could be used across additional syndromic and sporadic autoimmune diseases to develop an atlas of linear B cell autoantigens.

## Results and discussion

### Design and implementation of a scaled PhIP-seq protocol

The ability to process large numbers of patient samples for PhIP-seq in a highly uniform manner has several important benefits, including reduction of batch effects between samples from the same cohorts as well as between disease and control cohorts, detection of lower-frequency autoantigens, and the ability to simultaneously include large numbers of control samples.

In creating a scaled protocol, we searched for a method that would allow 600–800 samples to be run fully in parallel to reduce any batch or plate-to-plate variability. Thus, each wash or transfer step needed to be performed in rapid succession for all plates. We also prioritized reduction of any well-to-well contamination, particularly given that small, early contamination can amplify across subsequent rounds of immunoprecipitation. Finally, we required our protocol to minimize consumable waste and maximize benchtop accessibility without robotics or other specialized equipment.

A benchtop vacuum-based protocol (rapid, consistent wash times) in deep-well 96-well filter plates with single-use foil seals (no well-to-well contamination) met our requirements (see schematic in *Figure 1A*). The data for APS1 samples on our moderate-throughput manual multichannel protocols were closely correlated with vacuum-based output, including identification of previously validated antigens within each sample (*Figure 1B*). Additional protocol improvements included: 3-D printing template for vacuum plate adaptors (for easy centrifugation and incubation steps); direct addition of protein A/G beads to *Escherichia coli* lysis without a preceding elution step; shortened lysis step by using square-well plates with semi-permeable membrane for aeration; and options for smaller volume and decreased reagent library preparation in both 96-well and 384-well formats. A detailed protocol, including custom part designs, for both high-throughput vacuum-based and moderate throughput multichannel-based protocols is available at protocols.io.

### Large control cohorts are critical for identifying disease-specific autoantibodies

Some assays for autoantigen discovery, such as protein arrays, are often used as a quantitative measure of how autoreactive an individual serum sample may be. In contrast, PhIP-seq is an enrichment-based assay in which binders are serially enriched and amplified. A practical limitation of this technique is that non-specific phage may also be amplified, in addition to a wide array of autoreactive, but non-disease-related peptides. We tested whether we could detect global differences between case and control cohorts as a measure of autoreactivity. As each APS1 patient is known to have multiple, high-affinity antibodies to self-proteins (*Fishman et al., 2017*; *Landegren et al., 2016*; *Meyer et al., 2016*; *Vazquez et al., 2020*) we reasoned that this would be an ideal cohort to determine whether a global autoreactive state was discernible. As expected, each individual sample exhibits a spectrum of enriched genes, regardless of disease status (*Figure 2A*), indicating that measures of simple enrichment are inadequate for discrimination of cases from controls.

We and others have shown that PhIP-seq can robustly detect disease-associated antigens by comparing antigen-specific signal between disease and control cohorts (*Larman et al., 2011*; *Mandel-Brehm et al., 2019*; *O'Donovan et al., 2020*; *Vazquez et al., 2020*). In this dataset, encompassing 186 control samples and 128 APS1 samples, we further evaluated the importance of control cohort size. We iteratively downsampled the number of healthy control samples in our dataset to 5, 10, 25, 50, 100, or 150 (out of n=186 total control samples). The number of apparent hits was determined in each condition, where a gene-level hit was called when the following criteria were met: (1) at least 10% of APS1 samples and less than two control samples with a Z-score >10, (2) no control sample exhibiting higher signal than the highest patient signal, and (3) a minimum of one strong positive

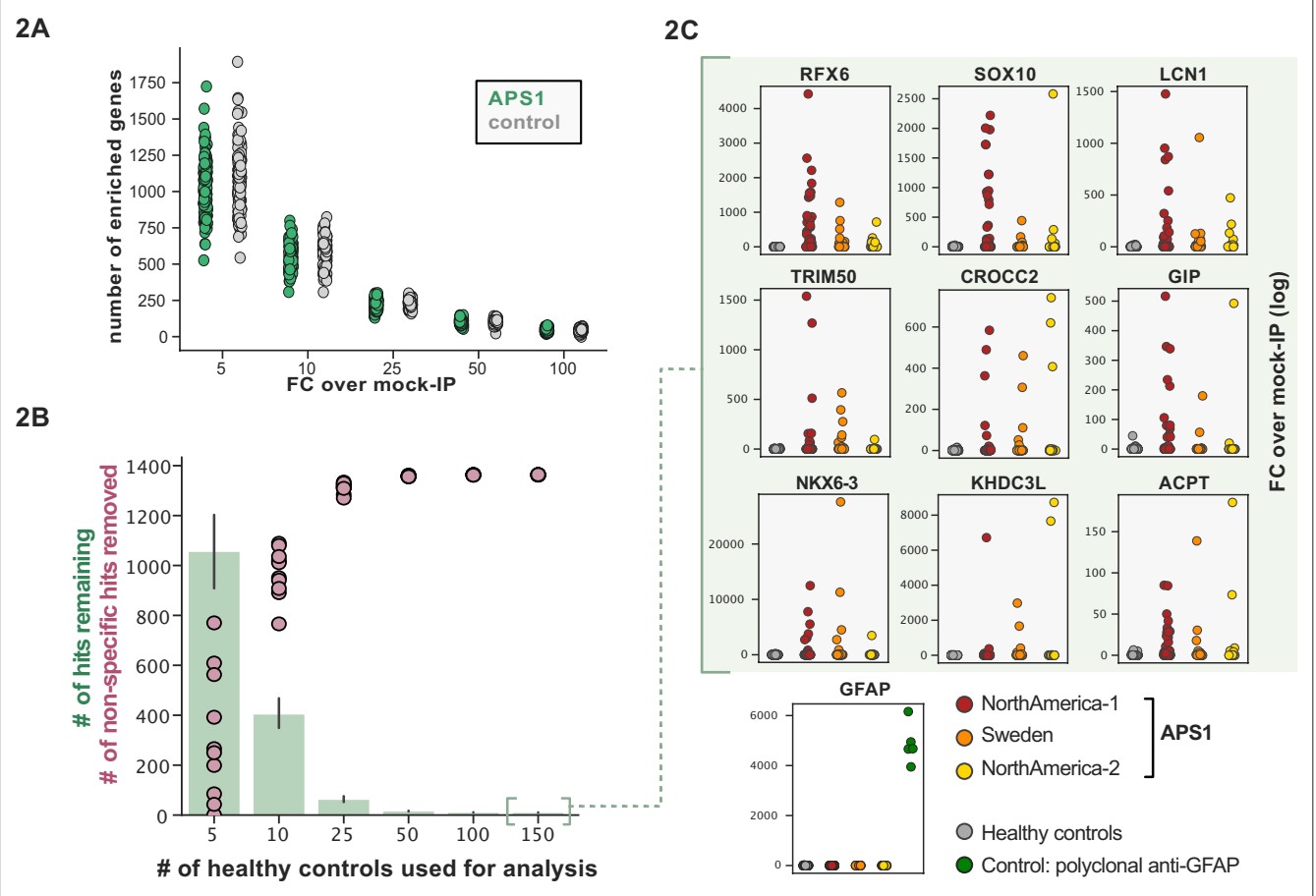

**Figure 2.** Application of scaled phage-immunoprecipitation sequencing to expanded APS1 and healthy control cohorts. (**A**) Number of hits per sample reaching 5, 10, 25, 50, and 100-fold enrichment relative to mock-IP samples. Each dot represents a single APS1 patient (green) or non-APS1 control (gray). (**B**) When looking for disease-specific hits, increasing the number of healthy controls results in fewer apparent hits and is therefore critical. Shared hits are defined as gene-level signal (>10-fold change over mock-IP) which is shared among 10% of APS1 samples (n=128), present in fewer than 2% of healthy controls, and with at least one APS1 sample with a high signal (FC of 50<). Random downsampling was performed 10 times for each healthy control bin. (**C**) Nine gene-level hits are present in 10%< of a combined three-group APS1 cohort. North-America-1, n=62; Sweden, n=40; North-America-2, n=26. Anti-GFAP control antibody (n=5) indicates that results are consistent across plates and exhibit no well-to-well contamination.

patient sample (50-fold enrichment over mock-IP). Genes that failed to meet these conditions were considered non-specific. Using these conservative criteria, a control set of 10 samples resulted in (on average) 404 apparent hits, while increasing the control set to 50 samples removed an additional 388 non-specific hits, leaving 16 apparent hits (*Figure 2B*). Further increasing the number of control samples to 150 samples had diminishing returns, although 4–5 more autoantigen candidates were removed as being non-specific, reducing the frequency of false positives, and ultimately leaving only ~1% of the original candidate list for further investigation. In sum, to improve downstream analysis aimed at detecting disease-associated hits, PhIP-seq experimental design should include a large and appropriate number of non-disease controls.

## Scaled PhIP-seq replicates and expands autoantigen repertoires across multiple independent cohorts of APS1

We previously identified and validated PhIP-seq hits based on shared positivity of a given hit among 3 (out of 39) or more patients with APS1 (*Vazquez et al., 2020*). While this enabled us to robustly detect frequently shared antigens within a small disease cohort, antigens with lower frequencies – or with low detection sensitivity – would likely fall below this conservative detection threshold.

To improve both sensitivity and specificity, we utilized scaled PhIP-seq to explore expanded cohorts of disease, including a much larger (n=128) APS1 cohort spanning two North American cohorts and

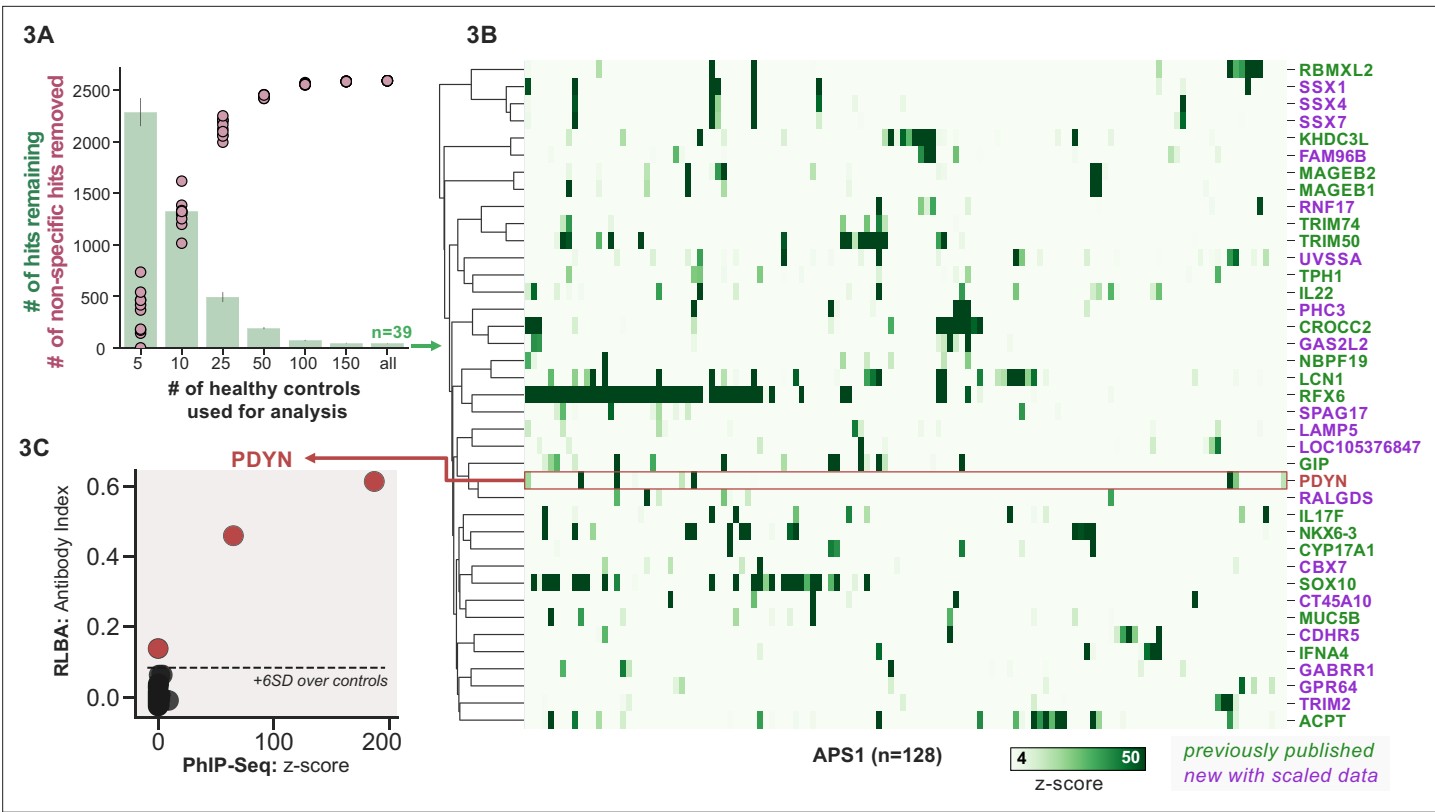

**Figure 3.** Replication and expansion of APS1 autoantigens across multiple cohorts using scaled phage-immunoprecipitation sequencing (PhIP-seq). (**A**) Increasing the number of healthy controls results in fewer apparent hits and is therefore critical. Shared hits are defined as gene-level signal (>10-fold change over mock-IP) which is shared among 4%< of APS1 samples (n=128), present in fewer than 2% of healthy controls, and with at least one APS1 sample with a high signal (FC of 50<). Random downsampling was performed 10 times for each healthy control bin. (**B**) 39 candidate hits present in 4%< of the APS1 cohort. (**C**) Rare, novel anti-PDYN autoantibodies validate at whole-protein level, with PhIP-seq and whole-protein RLBA data showing good concordance.

one Swedish cohort. All hits present in at least 10% of APS1 patients also spanned all three cohorts, thus further validating broad prevalence of antigens that were previously described by us (RFX6, ACPT, TRIM50, CROCC2, GIP, NKX6-3, KHDC3L) and others (SOX10, LCN1) (*Figure 2C*; *Fishman et al., 2017*; *Hedstrand et al., 2001*; *Vazquez et al., 2020*). At the gene level, we detected 39 candidate hits that were present in 6/128 (>4%) of APS1 and in 2 or fewer controls (2/186, 1%) (*Figure 3A and B*). As expected, the larger cohort yielded new candidate antigens that had not been detected previously. For example, PDYN is a secreted opioid peptide thought to be involved in regulation of addiction-related behaviors (*reviewed in Fricker et al., 2020*). PDYN was enriched by 6/128 (4.7%) of patient samples and subsequently was validated by Radioligand Binding Assay (RLBA) (*Figure 3C*). Indeed, this validated antigen was present in our previous investigations (*Vazquez et al., 2020*); however, it was enriched in too few samples to qualify for follow-up.

Other notable hits with relevant tissue-restricted expression were also observed. For example, SPAG17 is closely related to the known APS1 autoantigen SPAG16 and is expressed primarily in male germ cells and in the lung, with murine genetic mutations resulting in ciliary dyskinesis with pulmonary phenotypes (*Fishman et al., 2017*; *Teves et al., 2013*). Also potentially related to ciliary and/or pulmonary autoimmunity is CROCC2/Rootletin, a protein expressed in ciliated cells, which we previously observed, and now recognize at a high frequency across multiple cohorts (*Uhlén et al., 2015*; *Yang et al., 2002*). Similarly, GAS2L2 is a ciliary protein expressed in the airway, with genetic loss of function in mice resulting in impaired mucociliary clearance, and clustered closely with CROCC2 (*Bustamante-Marin et al., 2019*; *Uhlén et al., 2015*) in this dataset. These novel putative antigens together hint at potential mucociliary airway autoreactivity. CT45A10 and GPR64 are both proteins with expression restricted primarily to male gonadal tissues (*Uhlén et al., 2015*). GABRR1 is a GABA receptor expressed primarily in the central nervous system as well as on platelets (*Ge et al., 2006*;

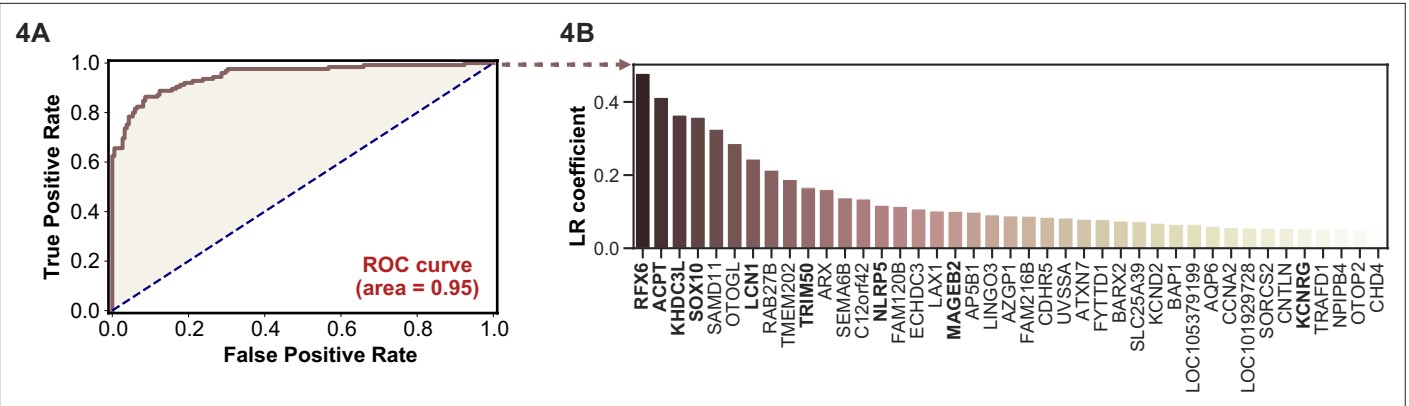

**Figure 4.** Logistic regression of phage-immunoprecipitation sequencing data enables APS1 disease prediction. (**A**) Receiver operating characteristic (ROC) curve for prediction of APS1 versus control disease status. (**B**) The highest logistic regression (LR) coefficients include known antigens RFX6, KHDC3L, and others.

*Zhu et al., 2019*), and TRIM2 is implicated in genetic disorders of demyelination within the peripheral nervous system, and therefore may be of interest to the chronic inflammatory demyelinating polyneuropathy phenotype that can be seen in some patients with APS1 (*Li et al., 2020*; *Valenzise et al., 2017*). In addition to our previously described intestinally expressed autoantigens RFX6, NKX6-3, and GIP, we also identify CDHR5, a transmembrane cadherin-family protein expressed on the enterocyte cell surface, as a putative autoantigen in APS1 (*Crawley et al., 2014*; *Uhlén et al., 2015*).

## APS1 disease prediction by machine learning

APS1 is a clinically heterogeneous disease, and it is also heterogeneous with respect to autoantibodies (*Ferre et al., 2016*; *Fishman et al., 2017*; *Landegren et al., 2016*; *Meyer et al., 2016*; *Vazquez et al., 2020*). Because PhIP-seq simultaneously interrogates autoreactivity to hundreds of thousands of peptides, we hypothesized that unsupervised machine learning techniques could be used to create a classifier that would distinguish APS1 cases from healthy controls. We applied a simple logistic regression classifier to our full gene-level APS1 (n=128) and control (n=186) datasets, resulting in excellent prediction of disease status (AUC = 0.95, *Figure 4A*) using fivefold cross-validation. Moreover, we found that the classification model was driven strongly by many of the previously identified autoantigens, including RFX6, KHDC3L, and others (*Figure 4A*), in addition to some targets that had not been previously examined (*Vazquez et al., 2020*). These results demonstrate that PhIP-seq autoreactive antigen enrichment profiles are amenable to machine learning techniques, and further suggest that such data could be used to derive diagnostic signatures with strong clinical predictive value.

## Antoantibody discovery in IPEX

IPEX syndrome is characterized by defective peripheral immune tolerance secondary to impaired T regulatory cell (Treg) function. In IPEX, peripheral tolerance rather than central tolerance is impaired, resulting in a phenotypic constellation of autoimmunity that partially overlaps with APS1 (*Bacchetta et al., 2006*; *Powell et al., 1982*). Notably, the majority of IPEX patients exhibit severe enteropathy, with early-onset severe diarrhea and failure to thrive, with many of these children harboring anti-enterocyte antibodies detected by indirect immunofluorescence (*Bacchetta et al., 2006*; *Gambineri et al., 2018*; *Powell et al., 1982*). We hypothesized that the same PhIP-seq approach that was successful for APS1 would also yield informative hits for IPEX. A total of 27 patient samples were analyzed using scaled PhIP-seq, and the data processed in the same manner as for APS1.

A handful of IPEX autoantibodies targeting antigens expressed in the intestine have been described, including harmonin (USH1C) and ANKS4B (*Eriksson et al., 2019*; *Kobayashi et al., 2011*). In our data, enrichment of USH1C was observed in two IPEX patients, and this signal was fully correlated with anti-ANKS5B as previously described (*Figure 5—figure supplement 1*; *Eriksson et al., 2019*).

Several novel putative autoantigens were observed that were shared among three or more IPEX patients (*Figure 5A*). Among these were several with expression restricted to the intestine, including

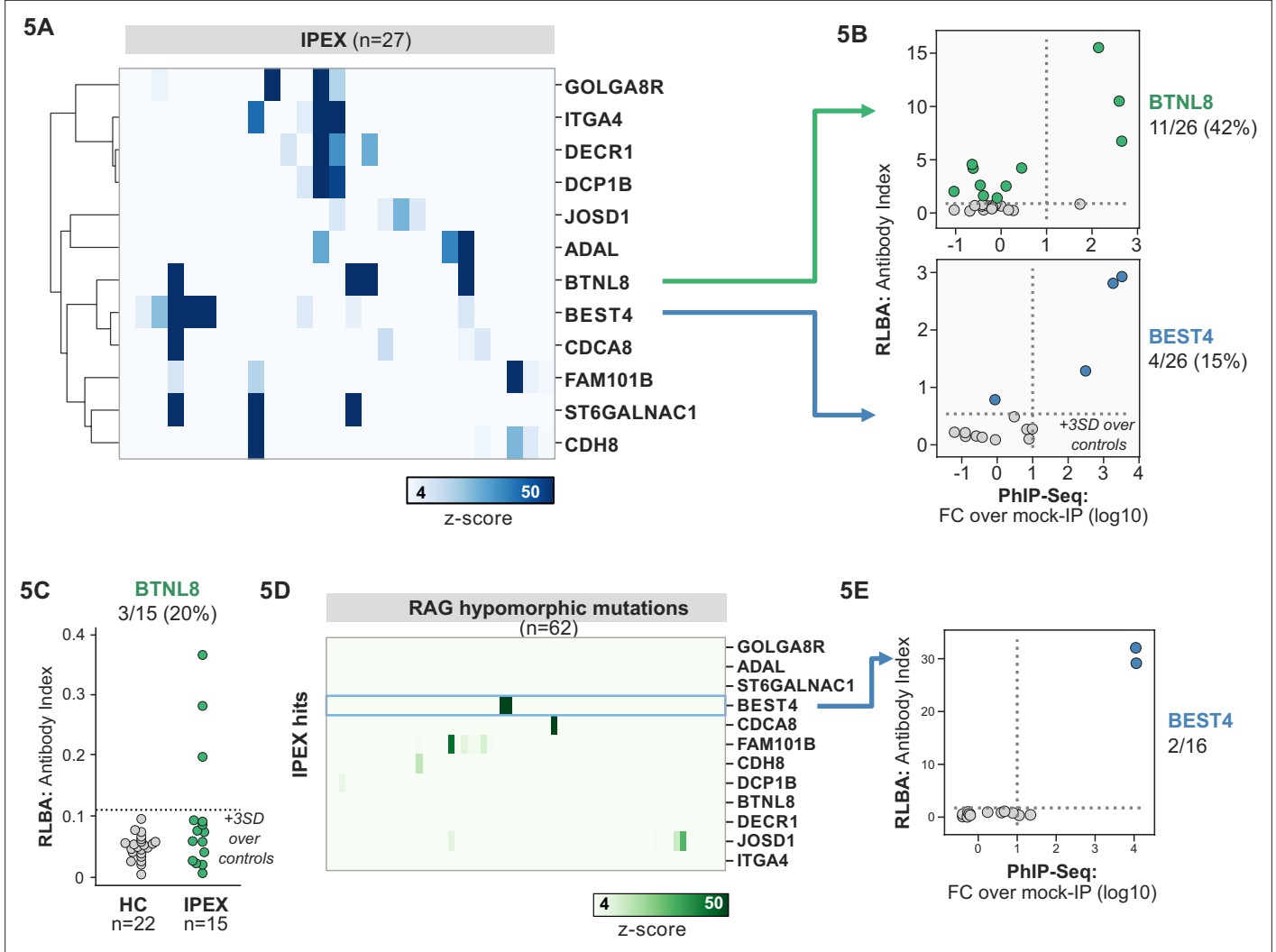

**Figure 5.** Phage-immunoprecipitation sequencing (PhIP-seq) screening in IPEX and RAG1/2 deficiency reveals novel, intestinally expressed autoantigens BEST4 and BTNL8. (**A**) PhIP-seq heatmap of most frequent shared antigens among IPEX, with color indicating z-score relative to a cohort of non-IPEX controls. (**B**) Radioligand binding assay for BTNL8 reveals additional anti-BTNL8 positive IPEX patients (top). Radioligand binding assay for BEST4 autoantibodies correlates well with PhIP-seq data (bottom). (**C**) Discovery of additional anti-BTNL8 positive individuals in an independent IPEX cohort (n=15) by radioligand binding assay; dotted line indicates mean of healthy controls +3 SD. (**D**) PhIP-seq screen of patients with hypomorphic mutations in *RAG1/2* reveals two patients with anti-BEST4 signal. (**E**) Orthogonal radioligand binding assay validation of anti-BEST4 antibodies in both PhIP-seq anti-BEST4 positive patients.

The online version of this article includes the following figure supplement(s) for figure 5:

**Figure supplement 1.** PhIP-seq application to a cohort of patients with hypomorphic *RAG1/2* mutations.

BEST4, a protein expressed by a specific subset of enterocytes, BTNL8, a butyrophilin-like molecule highly expressed in the gut epithelium, ST6GALNAC1, and ITGA4 (*Figure 5A*; *Mayassi et al., 2019*; *Schaum et al., 2018*; *Uhlén et al., 2015*). BEST4 and BTNL8 were selected for validation by whole protein immunoprecipitation. A total of 4/26 (15%) of IPEX patients were positive for anti-BEST4 autoantibodies (*Figure 5B*). In the case of BTLN8, orthogonal validation identified 11/26 (42%) of IPEX patients who were positive for anti-BTNL8 antibodies (*Figure 5B*). Of note, all patients with anti-BTNL8 and/or BEST4 antibodies also had clinical evidence of enteropathy (*Supplementary file 1*). In addition, to broaden the validity of our findings, we tested a second, independent cohort of IPEX patients for anti-BTNL8 antibodies and found 3/15 (20%) patients to be positive (*Figure 5C*). Taken together, these results suggest that anti-BEST4 and anti-BTNL8 are associated with IPEX enteropathy.

## Overlap of intestinal autoantigen BEST4 in the setting of hypomorphic RAG1/2 mutations

Hypomorphic *RAG1/2* mutations represent an additional and notoriously phenotypically heterogeneous form of monogenic immune dysregulation. Absent RAG complex activity leads to lack of peripheral T and B cells, therefore causing severe combined immunodeficiency (SCID). However, patients with hypomorphic *RAG1/2* have residual capacity to generate T and B cells. Depending on the severity of the defect, these patients can present with Omenn syndrome, atypical SCID (AS), or combined immune deficiency with granulomas and autoimmunity (CID-G/AI) (*Delmonte et al., 2018*; *Delmonte et al., 2020*). Autoimmune manifestations are particularly common in patients with AS and CID-G/AI. Cytopenias are the most frequent form of autoimmunity in patients with RAG deficiency, but cutaneous, neuromuscular, and intestinal manifestations have been also reported. While anti-cytokine antibodies have been described, other disease-associated autoantibody targets remain to be identified (*Delmonte et al., 2020*).

62 patients with hypomorphic RAG1/2 mutations were screened by PhIP-seq to assess for overlap with APS1 and IPEX antigens, as well as for novel autoantigen specificities. Minimal overlap was observed between RAG1/2 deficiency and APS1 and IPEX. However, two samples from RAG1/2 patients indicated the presence of anti-BEST4 antibodies (*Figure 5D*), which were confirmed through orthogonal validation using whole protein (*Figure 5E*). Both positive patients had CID-G/AI, indicating the presence of autoimmune features. Remarkably – given that enteropathy in the setting of hypomorphic RAG1/2 deficiency is rare – one of the two individuals harboring anti-BEST4 antibodies also had very early-onset inflammatory bowel disease.

Several other putative antigens among the larger RAG1/2 deficiency cohort were revealed. Nearly half of the cohort sera were enriched for peptides derived from ZNF365 (*Figure 5—figure supplement 1B*). ZNF365 is a protein associated with multiple autoimmune diseases, as evidenced by genome-wide association studies (GWAS) show that variants in ZNF365 are associated with Crohn's disease and autoimmune uveitis (*Haritunians et al., 2011*; *Hou et al., 2020*). Many patients also had evidence of autoantibodies targeting RELL2, a TNF receptor binding partner, and CEACAM3, a phagocyte receptor that recognizes human-specific pathogens and is important for opsonin-independent phagocytosis of bacteria (*Bonsignore et al., 2019*; *Moua et al., 2017*). Autoantibodies targeting these antigens could potentially play a role in the autoinflammation seen in certain cases of hypomorphic *RAG1/2* and/or increased susceptibility to particular infections and will require additional future follow-up.

## PhIP-seq identifies rare, shared candidate autoantigens in MIS-C

MIS-C leads to critical illness in ~70% of affected children and shares some common clinical features with KD, the most common cause of acquired pediatric heart disease in the United States. Despite hints for a role of abnormal adaptive immunity and autoantibodies in the pathogenesis of KD and MIS-C, the etiologies of both diseases remain enigmatic (*Feldstein et al., 2020*; *Newburger et al., 2016*). Recently, PhIP-seq has been deployed to explore COVID-19-associated MIS-C (*Gruber et al., 2020*). However, this study included only four healthy controls and nine MIS-C patients, and as our results have shown, removal of false-positive PhIP-seq hits requires the use of substantial numbers of unaffected controls (*Figure 2C*). Furthermore, these previously published hits lacked orthogonal validation. Therefore, we sought to examine an MIS-C cohort in light of these results, as well as to explore for possible autoantibody overlap between KD and MIS-C.

First, 20 MIS-C subjects, 20 pediatric febrile controls, and 20 COVID-19 controls were examined by PhIP-seq, each of which was compared to a cohort of adult healthy controls (n=87). No evidence for specific enrichment was observed for any of the previously reported candidate antigens that overlapped with our PhIP-seq library (*Figure 6A*). Methodologic and sample differences could, in part, account for differences in our results; however, these results suggest that PhIP-seq hits should be subjected to external replication and/or validation. Additionally, MIS-C patients are treated with IVIG, which has abundant low-affinity, low-avidity autoantibodies, so caution must be exercised in the timing of sample collection (*Burbelo et al., 2021*; *Sacco et al., 2021*). While it remains unclear whether high-affinity, novel autoantibodies are likely to be present in IVIG, the majority of our MIS-C samples are confirmed to be collected pre-IVIG. Though several of our samples are of unknown IVIG

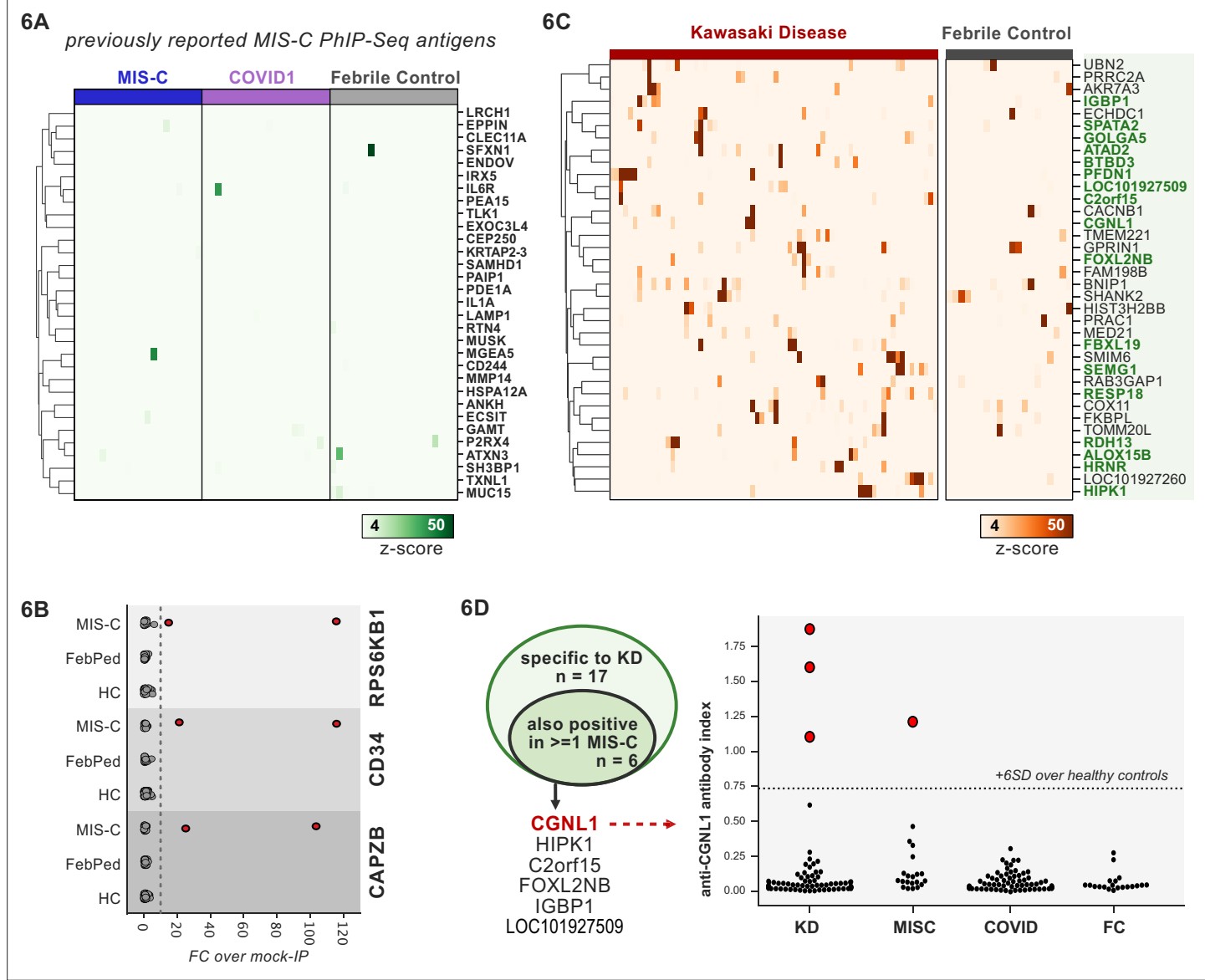

**Figure 6.** Phage-immunoprecipitation sequencing (PhIP-seq) screening of multisystem inflammatory syndrome in children (MIS-C) and Kawasaki disease (KD) cohorts. (**A**) Heatmap of signal for putative hits from *Gruber et al., 2020*, among MIS-C, adult COVID-19 controls, and pediatric febrile controls (each n=20). (**B**) Only rare, shared PhIP-seq signals were found among n=20 MIS-C patients. (**C**) Heatmap of putative antigens in a cohort of n=70 KD patients. Hits that are specific to KD and are not found among n=20 febrile controls, are highlighted in green. (**D**) A small number of rare putative antigens are shared between KD and MIS-C (left), with radioligand binding assay confirmation of antibody reactivity to whole protein form of CGNL1 in three KD patients and one MIS-C patient (right).

status, each of the candidate autoantibodies discussed below is present in at least one sample known to be IVIG-free.

Analysis of our MIS-C cohort for shared candidate hits yielded only three candidate hits, each in 2/20 patient sera. These were CD34, RPS6KB1, and CAPZB (*Figure 6B*). While these targets may be of interest, disease-association remains uncertain. These results suggest that a much larger MIS-C cohort, controlled by an equally large set of healthy controls, will be required to detect rare, shared antigens with confidence by PhIP-seq.

## PhIP-seq screen of a cohort of KD patients

To screen for possible KD-specific autoantibodies, we analyzed a large cohort of 70 KD subjects by PhIP-seq. KD patients are also often treated with IVIG, so care was taken to ensure that each of these

samples was collected prior to IVIG administration. Using the same hit selection criteria as previously, we detected 25 shared hits among 3 or more of the 70 KD samples, which were specific to KD relative to adult healthy controls (*Figure 6C*). Of these 25 shared KD hits, 17 were absent from additional control groups including the febrile pediatric patients. Each of these hits was present in only a small subset of KD samples, suggesting significant heterogeneity among samples.

Some of the candidate antigens have possible connections to the systemic inflammation seen in KD, including SPATA2 (6/70, 8.6%) and ALOX5B (4/70, 5.7%). SPATA2 is a protein known to regulate the TNF receptor response, with murine knockout of SPATA2 resulting in increased activation of NFkB and MAPK (*Schlicher and Maurer, 2017*). Similarly, ALOX15 family of lipoxygenases is known to be responsive to Th2-induced anti-inflammatory cytokine IL-4 and IL-13 in human macrophages and thereby likely plays a role in suppressing inflammatory responses, and polymorphisms have been linked to the development of coronary artery disease (*Snodgrass and Brüne, 2019*; *Wuest et al., 2014*).

Other KD candidate autoantigens exhibited tissue expression patterns that suggest possible relationships to sub-phenotypes of KD. For example, pancreatitis and psoriasis have been reported as rare complications of KD (*Asano et al., 2005*; *Haddock et al., 2016*). We found that 7/70 of the KD patients (10%) have increased signal for autoantibodies targeting FBXL19, a protein with associations with both acute pancreatitis (*Ma et al., 2020*) and psoriasis (*Stuart et al., 2010*) and 10/70 for pancreas-expressed protein RESP18 (*Zhang et al., 2007*).

Taken together, our dataset did not uncover the presence of common autoantibodies in KD or MIS-C. Nonetheless, our findings leave open the possibility of lower-frequency disease-associated autoantibodies, and future studies with increased cohort scaling, diverse representation of clinical subphenotypes, and high-sensitivity follow-up assays may shed further light on the role of autoantibodies in KD.

## Autoantigen overlap between KD and MIS-C

Given the partial clinical overlap of KD with MIS-C, we also searched for the low-frequency, shared KD hits within MIS-C and found that 6 of the 17 KD hits were present in one or more MIS-C samples (*Figure 6D*). Of these, CGNL1 was of particular interest given the very high enrichment values in patients and absence of signal from all controls, as well as its anatomic expression pattern. CGNL1 is

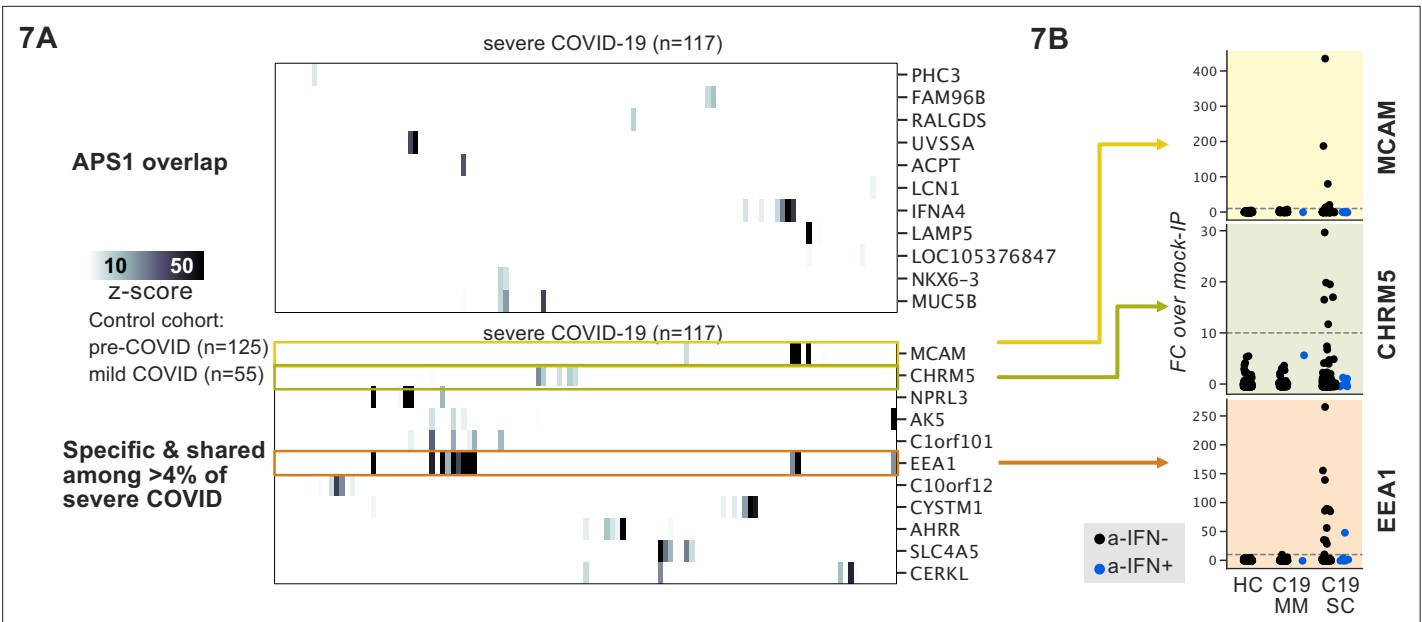

**Figure 7.** Phage-immunoprecipitation sequencing screening in severe forms of COVID-19 reveals putative novel autoantigens, including EEA1. (**A**) Screening of patients with severe COVID-19 pneumonia shows little overlap with APS1 but enables discovery of possible novel disease-associated autoantigens including EEA1. (**B**) Putative novel antigens EEA1, CHRM5, and MCAM are primarily found in anti-IFN-negative patients, suggesting the possibility of other frequent, independent disease-associated antibodies in severe COVID-19.

an endothelial junction protein and is highly expressed in the cardiac endothelium (*Chrifi et al., 2017*; *Schaum et al., 2018*). We confirmed anti-CGNL1 autoantibodies using a radioligand binding assay with whole CGNL1 protein (*Figure 6D*). We also identified an additional positive KD patient that had not been detected by PhIP-seq. These data suggest that anti-CGNL1 antibodies, while rare, may be associated with KD and/or MIS-C.

## Application of scaled PhIP-seq to severe COVID-19

Recently, it was reported that over 10% of severe COVID-19 pneumonia is characterized by the presence of anti-Type I IFN autoantibodies, a specificity that overlaps with APS1 (*Bastard et al., 2020*; *Bastard et al., 2021b*; *Bastard et al., 2021a*; *Meager et al., 2006*; *Meyer et al., 2016*; *van der Wijst et al., 2021*). We therefore looked for possible overlap of additional antigens between APS1 and COVID-19. As expected, we had low sensitivity for the known anti-type I IFN autoantibodies by PhIP-seq, likely due to the conformational nature of these antigens; however, this cohort had been previously assessed for anti-Type I IFN antibodies by other techniques (*Bastard et al., 2020*; *Bastard et al., 2021c*). We did not detect substantial overlap of any of the other antigens that were found in 5% or more of APS1 samples, suggesting that autoantibody commonalities between APS1 and severe COVID-19 may be limited (*Figure 7A*).

PhIP-seq was then used to investigate patients with severe to critical COVID-19 pneumonia (n=117) relative to patients with mild to moderate COVID-19 (n=55) and non-COVID-19 healthy controls (n=125) (*Figure 7A*). A small number of putative autoantigens were identified in four or more of the severe COVID-19 group (>4%), but not within our control group, including MCAM, CHRM5, and EEA1. Of these, only EEA1, an early endosomal protein, had a frequency of >10% (*Mu et al., 1995*). Notably, the set of anti-EEA1 positive patients was almost fully distinct from the anti-Type I IFN antibody positive group, with only one patient shared between the groups (*Figure 7B*). Given the importance of the finding of autoantibodies to Type 1 IFNs in >15% of critical COVID-19 pneumonia, our data suggest that anti-EEA1 antibodies may also have predictive potential worth further investigation.

## Discussion

PhIP-seq is a powerful tool for antigen discovery due to its throughput and scalability. Continually declining costs in sequencing paired with scaled protocols such as the ones presented here result in a low per-sample cost and experimental time, and declining costs of custom oligonucleotide based-libraries allow for extensive adaptation (*Román-Meléndez et al., 2021*; *Vogl et al., 2021*). Yet, as the technology is relatively new, increased discussion around best practices in experimental design, methodology, and data interpretation would be beneficial. In this work, we contribute to these efforts by presenting an accessible, carefully documented scaled lab protocol. We anticipate that scaling and low cost will facilitate cross-platform comparisons of autoimmune sera, enabling a more systematic understanding of each platform's abilities and shortcomings, as well as the best approaches to combine platforms for comprehensive autoantibody profiling, including fixed protein arrays, yeast display (REAP), and PhIP-seq (*Wang et al., 2021*).

Our scaled data reinforces the importance of including large control cohorts to improve specificity and orthogonal validations to evaluate sensitivity. The absence of appropriately sized control cohorts can result in false-positive associations between enriched protein sequences and disease which can lead to misinterpretation. Furthermore, PhIP-seq is optimized for linear antigens and thus is inherently less robust for detection of confirmational or modified epitopes. For example, our previous work highlights that known anti-IFN and anti-GAD65 antibodies can be detected only in a handful of APS1 samples by PhIP-seq, while orthogonal assays using whole conformation protein demonstrate increased sensitivity (*Vazquez et al., 2020*). Similarly, the increased sensitivity of whole protein validation relative to PhIP-seq for BTNL8 autoantibodies highlights the need for rapid and sensitive secondary assays to confirm or even increase the importance of a given candidate antigen. These principles may be extended to other related PhIP-seq modalities (*Mina et al., 2019*; *Román-Meléndez et al., 2021*; *Vogl et al., 2021*).

Several novel APS1 antigens were identified in this work by virtue of the large cohort size and large control set, enabling the detection of low-frequency hits such as PDYN. PDYN is a secreted opioid precursor of the central nervous system, and while PDYN-knockout mice are largely phenotypically

normal, they do experience hyperalgesia and altered anxiety-related behaviors relative to wild-type mice (*McLaughlin et al., 2003*; *Sharifi et al., 2001*; *Wang et al., 2001*). Future studies will be required to determine if anti-PDYN antibodies may themselves mediate these phenotypes. In addition to uncovering lower-frequency autoantigens, antigen overlap between APS1 and other syndromic autoimmune diseases, including IPEX and RAG1/2 deficiency, were evaluated. Detection of common APS1 antigens, such as RFX6 and KHDC3L, was rare among other disease contexts, suggesting APS1 displays limited sharing with respect to other multiorgan autoimmune syndromes, including IPEX and RAG deficiency.

In contrast, antigen overlap between IPEX and RAG-hypomorphic patients was detected in the form of anti-BEST4 antibodies. BEST4 is a well-characterized, cell-surface intestinal ion channel (*Qu and Hartzell, 2008*). Recently, BEST4 has become a standard and specific marker for a subset of enterocytes in the duodenum and colon, including CFTR+ enterocytes, which are involved in fluid homeostasis (*Busslinger et al., 2021*; *Elmentaite et al., 2021*; *Smillie et al., 2019*). The presence of an IPEX antigen BEST4 within CID-G/AI suggests a possible etiologic link between the two diseases, possibly relating to Treg dysfunction. Furthermore, given that BEST4 autoantibodies were found across two distinct etiologies of IBD, in addition to the intestinal localization of BEST4 expression, future experiments specifically searching for the presence of anti-BEST4 antibodies in IBD and other forms of autoimmune enteropathy are warranted.

Another novel IPEX antigen, BTNL8, is also of particular interest given its high frequency in IPEX (11/26) as well as its biological functions. BTN and BTNL-family members belong to the family of B7 co-stimulatory receptors known to modulate T-cell responses, with structural similarity to CD80 and PD-L1 (*Chapoval et al., 2013*; *Rhodes et al., 2016*). Broadly, BTNL-family members are thought to participate primarily in regulation of gamma-delta T cells, with BTNL8/BTNL3 implicated in gut epithelial immune homeostasis (*Di Marco Barros et al., 2016*; *Chapoval et al., 2013*). Given the cell-surface expression pattern of BTNL8, it is conceivable that antibodies to these proteins could play a functional role in immune checkpoints, rather than only the bystander role implicated for most auto-antibodies to intracellular antigens. Interestingly, recent studies in patients with celiac disease have demonstrated a reduction in BTNL3/BTNL8 expression following acute episodes of inflammation, with an associated loss of the physiological normal gamma/delta T-cell subset of gut intraepithelial lymphocytes (*Mayassi et al., 2019*).

While some disease cohorts yield strongly enriched common antigens, such as BTNL8 and BEST4, other cohorts appear heterogeneous. Such was the case for MIS-C and KD, where our results suggested only rare, shared putative antigens. Nonetheless, a number of observations point to a possible role for abnormal adaptive immunity and autoantibodies (reviewed in *Sakurai, 2019*). Both diseases respond to varying degrees to IVIG therapy, and genetic studies suggest some children with polymorphisms in genes involving B-cell and antibody function can be predisposed to developing KD (reviewed in *Onouchi, 2018*). Autoantibodies targeting clinically relevant tissues such as the heart or endothelium have been detected in KD, though their role in disease pathogenesis is uncertain (*Cunningham et al., 1999*; *Fujieda et al., 1997*; *Grunebaum et al., 2002*). Although anti-CGNL1 antibodies were only found at low frequency, it is possible that these antibodies may represent a subset of specificities within anti-endothelial cell antibodies, given the endothelial expression of CGNL1 as well as its impli-cations in cardiovascular disease (*Chrifi et al., 2017*). Particularly in the setting of MIS-C, expanded cohorts both to further validate CGNL1 as well as for as-of-yet undescribed antigen specificities may add additional information to these otherwise poorly understood diseases.

There is increasing evidence for a role for autoantibodies in other acquired disease states that have not been classified per se as autoimmune diseases, including severe to critical COVID-19 pneumonia (*Bastard et al., 2020*; *Bastard et al., 2021c*; *van der Wijst et al., 2021*). Peptides derived from EEA1 were enriched in 11% of patients with severe COVID-19 pneumonia but not in patients with mild COVID-19. Interestingly, similarly to anti-IFN antibodies, anti-EEA1 antibodies have also previously been reported in the setting of systemic lupus erythematosus (*Selak et al., 2000*; *Stinton et al., 2004*; *Waite et al., 1998*). While intriguing, future studies will be needed to determine whether the speci-ficity of EEA1 autoantibodies to severe-critical COVID-19 can be replicated in independent cohorts, as well as in orthogonal assays.

This work suggests that PhIP-seq data may have predictive or diagnostic value when the cohorts are sufficiently large. Implementation of a logistic regression machine learning model with cross-validation

successfully discriminates APS1 samples from healthy control samples with high sensitivity and specificity (*Figure 4A*). Thus, scaled PhIP-seq represents an avenue to not only discover and compare novel autoantigens within and across cohorts of autoimmune disease, but also suggests that machine learning techniques will be effective tools for determining, without supervision, the underlying proteins that drive successful classification using this data type. This, in turn, may have diagnostic or prognostic value in a wide spectrum of immune dysregulatory contexts.

Finally, we emphasize that to broadly understand autoantibody profiles, full PhIP-seq data availability is critical for enabling expanded analysis and comparison across datasets, cohorts, and research groups. Importantly, groups with expertise in specific disease areas may add value by re-analyzing and evaluating candidate hits using additional metrics for hit prioritization, including increased weighting of orthogonal expression and/or genetic data. Full PhIP-seq data for all cohorts presented here will be linked to this publication and will be available for download at Dryad.

## Materials and methods
### PhIP-seq protocols
All PhIP-seq protocols described in detail are available at protocols.io, located at the links below.

Vacuum-based scaled: https://www.protocols.io/view/scaled-high-throughput-vacuum-phip-protocol-ewov1459kvr2/v1

Multichannel-based scaled: https://www.protocols.io/view/scaled-moderate-throughput-multichannel-phip-proto-8epv5zp6dv1b/v1

Library Preparation: https://www.protocols.io/view/phage-display-library-prep-method-rm7vz3945gx1/v1

### PhIP-seq data alignment and normalization
Fastq files were aligned at the level of amino acids using RAPSearch2 (*Zhao et al., 2012*). For gene-level analysis, all peptide counts mapping to the same gene were summed. 0.5 reads were added to all genes, and raw reads were normalized by converting to percentage of total reads per sample. Fold change (FC) over mock-IP was calculated on a gene-by-gene basis by dividing sample read percentage by mean read percentage in corresponding AG bead-only samples. Z-scores were calculated using FC values; for each disease sample by using all corresponding healthy controls, and for each healthy control samples by using all other healthy controls. The positive threshold used for detection of shared candidate antigens was a Z-score $\geq 10$. Shared hits were then determined by positive rate in the specified percentage of patient samples and under a specified percentage (<2% unless otherwise specified) in controls. In addition, at least one positive sample was required to have a minimum FC of 50 or above. Finally, no candidate antigens were allowed where any positive control samples signal fell above the highest patient sample.

### Radioligand binding assay
Radioligand binding assay was performed as described in *Vazquez et al., 2020* for each gene of interest (GOI). Briefly, GOI-myc-flag construct was used to in vitro transcribe and translate [35 S]-methionine-labeled protein, which was subsequently column purified, immunoprecipitated with patient serum on Sephadex protein A/G beads, and counts per million (cpm) were read out on a 96-well liquid scintillation reader (MicroBeta Trilux, Perkin Elmer). Constructs: PDYN (Origene, #RC205331), BEST4 (Origene, #RC211033), BTNL8(Origene, #RC215370), and CGNL1 (Origene, #RC223121). Anti-MYC (Cell Signaling Technologies, #2272 S) and/or anti-FLAG (Cell Signaling Technologies, #1493 S) antibodies were used as positive controls for immunoprecipitations.

### Statistics
Statistics for radioligand binding assay were performed as described in *Vazquez et al., 2020*. Briefly, the antibody index for each sample was calculated as: (sample value – mean blank value)/(positive control antibody value – mean blank value).

We applied a logistic regression classifier on log-transformed PhIP-seq RPK values from APS1 patients (n=128) versus healthy controls (n=186) using the scikit-learn package (*Pedregosa et al., 2011*) with a liblinear solver and L1 regularization. The model was evaluated with fivefold cross-validation.

### Ethics/human subjects research

Available clinical metadata varies across cohorts; all available clinical data is included in *Supplementary file 1*.

### APS1

North America - 1: All patient cohort data was collected and evaluated at the NIH, and all APECED/APS1 patients were enrolled in a research study protocol approved by the NIH Institutional Review Board Committee and provided with written informed consent for study participation (protocol #11-I-0187, NCT01386437). All NIH patients gave consent for passive use of their medical record for research purposes. The majority of this human cohort data was previously published *Ferre et al., 2016*; *Ferré et al., 2019*.

Sweden: Serum samples were obtained from Finnish and Swedish patients with APS1. All individuals met the clinical diagnostic criteria for APS1, requiring two of the hallmark components: chronic mucocutaneous candidiasis, hypoparathyroidism, and adrenal insufficiency, or at least one of the hallmark components in siblings or children of APS1 patients. The project was approved by local ethical boards and was performed in accordance with the declarations of Helsinki. All patients and healthy subjects had given their informed consent for participation.

North America - 2: All patients underwent informed consent with research study protocols approved by the UCSF Human Research Protection Program (IRB# 10-02467).

### RAG1/2 deficiency

Patients with RAG1/2 deficiency were enrolled in research study protocols approved by the NIAID, NIH Institutional Review Board (protocols 05-I0213, 06-I-0015, NCT03394053, and NCT03610802). Peripheral blood samples were obtained upon written informed consent.

### IPEX

IPEX patient cohort data was collected and evaluated at Seattle Children's Hospital under an IRB-approved protocol. Clinical data on this human cohort was previously published by *Gambineri et al., 2018*. For the independent validation cohort (*Figure 5C*), all patient cohort data was collected and evaluated at Stanford under an IRB-approved protocol and was previously published by *Narula et al., 2022*.

### KD/MIS-C/febrile controls

UCSD cohort (MIS-C/KD/febrile controls): The study was reviewed and approved by the institutional review board at the University of California, San Diego. Written informed consent from the parents or legal guardians and assent from patients were obtained as appropriate.

Rockefeller cohort (MIS-C): All individuals were recruited according to protocols approved by local Institutional Review Boards.

### Severe COVID-19

All COVID-19 (non-MIS-C) patients were collected between March 01, 2020 and July 21, 2020 and had positive results by SARS-CoV-2 RT-PCR in nasopharyngeal swabs. ZSFG remnant specimens (institutional review board [IRB] number 20-30387) were approved by the IRB of the University of California, San Francisco. The committee judged that written consent was not required for use of remnant specimens.

### Adult healthy controls

New York Blood Center & Vitalant Research Institute: Healthy, pre-COVID control plasma were obtained as deidentified samples. These samples were part of retention tubes collected at the time of blood donations from volunteer donors who provided informed consent for their samples to be used for research.

Rockefeller cohort: All individuals were recruited according to protocols approved by local Institutional Review Boards.

## Acknowledgements

We thank members of the DeRisi lab, members of the Anderson lab, Chukwuka Didigu, and Joseph M Replogle for helpful discussions. We thank the New York Blood Center and Vitalant Research Institute for providing deidentified healthy control plasma. The Laboratory of Human Genetics of Infectious Diseases is supported by the Howard Hughes Medical Institute, the Rockefeller University, the St. Giles Foundation, the National Institutes of Health (NIH) (R01AI088364 and R01AI163029), the National Center for Advancing Translational Sciences (NCATS), NIH Clinical and Translational Science Award (CTSA) program (UL1TR001866), the Fisher Center for Alzheimer's Research Foundation, the Meyer Foundation, the JPB Foundation, the French National Research Agency (ANR) under the "Investments for the Future" program (ANR-10-IAHU-01), the Integrative Biology of Emerging Infectious Diseases Laboratory of Excellence (ANR-10-LABX-62-IBEID), the French Foundation for Medical Research (FRM) (EQU201903007798), ANRS Nord-Sud (ANRS-COV05), ANR grants GENVIR (ANR-20-CE93-003), AABIFNCOV (ANR-20-CO11-0001), and GenMIS-C (ANR-21-COVR-0039), the European Union's Horizon 2020 research and innovation programme under grant agreement no. 824110 (EASI-Genomics), the Square Foundation, Grandir - Fonds de solidarité pour l'enfance, the SCOR Corporate Foundation for Science, Fondation du Souffle, Institut National de la Santé et de la Recherche Médicale (INSERM), REACTing-INSERM, and the University of Paris.

## Additional information

### Funding

| Funder | Grant reference number | Author |
| --- | --- | --- |
| National Institute of Allergy and Infectious Diseases | 5P01AI118688 | Mark S Anderson |
| National Institute of Allergy and Infectious Diseases | 1ZIAAI001175 | Michail S Lionakis |
| National Institute of Diabetes and Digestive and Kidney Diseases | 1F30DK123915 | Sara E Vazquez |
| Chan Zuckerberg Biohub | | Joseph L DeRisi |
| Parker Institute for Cancer Immunotherapy | | Mark S Anderson |
| Juvenile Diabetes Research Foundation United States of America | | Mark S Anderson |
| Helmsley Charitable Trust | | Mark S Anderson |
| National Institute of General Medical Sciences | 5T32GM007618 | Mark S Anderson |
| American Diabetes Association | 1-19-PDF-131 | Zoe Quandt |
| UCSF-CTSI TL1 | TR001871 | Zoe Quandt |
| Division of Intramural Research, National Institute of Allergy and Infectious Diseases | 1 ZIA AI001222 | Luigi D Notarangelo |
| Eunice Kennedy Shriver National Institute of Child Health and Human Development | 1R61HD105590 | Adriana Tremoulet |
| Laboratory of Human Genetics of Infectious Diseases | | Jean-Laurent Casanova |

| Funder | Grant reference number | Author |
|--------|------------------------|--------|
| FRM | EA20170638020 | Paul Bastard |
| Imagine Institute | MD-PhD program | Paul Bastard |

The funders had no role in study design, data collection and interpretation, or the decision to submit the work for publication.

## Author contributions

Sara E Vazquez, Conceptualization, Data curation, Formal analysis, Funding acquisition, Validation, Investigation, Visualization, Methodology, Writing – original draft, Protocol development and documentation; Sabrina A Mann, Data curation, Validation, Investigation, Writing – review and editing, Protocol development and documentation; Aaron Bodansky, Data curation, Formal analysis, Validation, Investigation, Writing – original draft, Writing – review and editing; Andrew F Kung, Software, Formal analysis, Visualization, Methodology, Writing – review and editing; Zoe Quandt, Elise MN Ferré, Nils Landegren, Daniel Eriksson, Paul Bastard, Shen-Ying Zhang, Adriana Tremoulet, Kara Lynch, Luigi D Notarangelo, Jane C Burns, Data curation, Writing – review and editing; Jamin Liu, Michael R Wilson, Methodology, Writing – review and editing; Anthea Mitchell, David Yu, Chung-Yu Wang, Brenda Miao, Olle Kämpe, Validation; Irina Proekt, Data curation, Validation; Caleigh Mandel-Brehm, Writing – review and editing; Gavin Sowa, Conceptualization, Resources, Supervision, Funding acquisition, Methodology, Project administration, Writing – review and editing; Kelsey Zorn, Data curation, Validation, Investigation, Project administration, Writing – review and editing; Alice Y Chan, Writing – original draft, Writing – review and editing; Veronica M Tagi, Resources, Data curation; Chisato Shimizu, Data curation, Software, Formal analysis, Visualization, Methodology, Writing – review and editing; Kerry Dobbs, Data curation, Methodology, Writing – review and editing; Ottavia M Delmonte, Data curation, Project administration; Rosa Bacchetta, Resources, Data curation, Funding acquisition; Jean-Laurent Casanova, Michail S Lionakis, Troy R Torgerson, Data curation; Mark S Anderson, Conceptualization, Resources, Data curation, Supervision, Funding acquisition, Investigation, Project administration, Writing – review and editing; Joseph L DeRisi, Conceptualization, Resources, Software, Supervision, Funding acquisition, Investigation, Visualization, Project administration, Writing – review and editing

## Author ORCIDs

Sara E Vazquez http://orcid.org/0000-0002-0601-7001
Sabrina A Mann http://orcid.org/0000-0002-4970-1073
Aaron Bodansky http://orcid.org/0000-0001-8943-8233
Nils Landegren http://orcid.org/0000-0002-6163-9540
Daniel Eriksson http://orcid.org/0000-0001-5473-3312
Brenda Miao http://orcid.org/0000-0002-3393-9837
Gavin Sowa http://orcid.org/0000-0002-2089-8116
Michael R Wilson http://orcid.org/0000-0002-8705-5084
Olle Kämpe http://orcid.org/0000-0001-6091-9914
Kerry Dobbs http://orcid.org/0000-0002-3432-3137
Troy R Torgerson http://orcid.org/0000-0003-3489-5036
Mark S Anderson http://orcid.org/0000-0002-3093-4758
Joseph L DeRisi http://orcid.org/0000-0002-4611-9205

## Ethics

Human subjects: Detailed information on consent, where applicable, is available in the methods section of the manuscript.

## Decision letter and Author response

Decision letter https://doi.org/10.7554/eLife.78550.sa1
Author response https://doi.org/10.7554/eLife.78550.sa2

## Additional files

### Supplementary files
• Supplementary file 1. Clinical metadata.
• Transparent reporting form

### Data availability
Full PhIP-Seq data for all cohorts presented is available for download at Dryad at https://doi.org/10.5061/dryad.qfttdz0k4. All available deidentified clinical data for this study is available in Supplementary file 1.

The following dataset was generated:

| Author(s) | Year | Dataset title | Dataset URL | Database and Identifier |
|-----------|------|---------------|-------------|-------------------------|
| Vazquez SE | 2022 | Autoantibody discovery across monogenic, acquired, and COVID19-associated autoimmunity with scalable PhIP-Seq | https://doi.org/10.5061/dryad.qfttdz0k4 | Dryad Digital Repository, 10.5061/dryad.qfttdz0k4 |

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
