## [Editor Report]

This is an important paper that is methodologically compelling. The work presents a series of enhancements to the PhIP-seq method of autoantibody discovery, improving scaling to larger cohorts and control populations, and increasing the ability to discover disease-specific immune responses. The approach is used to discover a novel, frequent autoantibody response (BTNL8) in IPEX patients, and will be an accessible approach to investigate the presence and specificity of autoantibodies in diseases where these have been difficult to define.

---

## [Decision Letter]

**Decision letter after peer review:**

Thank you for submitting your article "Autoantibody discovery across monogenic, acquired, and COVID19-associated autoimmunity with scalable PhIP-Seq" for consideration by *eLife*. Your article has been reviewed by 3 peer reviewers, one of whom is a member of our Board of Reviewing Editors, and the evaluation has been overseen by Satyajit Rath as the Senior Editor. The following individual involved in review of your submission has agreed to reveal their identity: Shiv Pillai (Reviewer #1).

Essential revisions:

1. While all the reviewers recognized that the methodological improvements described have value, there was consensus that they alone do not meet the bar of "substantial developments" or "significant improvements" that a Research Advance require. To move forward there would need to be some application of the new method that reveals important information about an autoimmune disease or an immunological mechanism.

*Reviewer #1 (Recommendations for the authors):*

This manuscript may have had two goals – one being to develop methodology for more accurate comparison of autoantibody data across many cohorts with large numbers of controls and the other to gain some information on biological/biomedical relevance.

The authors should more lucidly explain how their alterations of the PhIP-seq approach are more than incremental.

One of the biological findings of possible note was the shared hits between KD and MIS-C. Perhaps it offers one way in which the disease relevance of this manuscript could be strengthened. This part could be expanded on a bit with more clinical information examining whether the MIS-C patients with antibodies to CGNL1 had more obvious cardiac findings compared to other MIS-C patients and perhaps consider tapping into other MIS-C cohorts for whom clinical information is also available.

*Reviewer #3 (Recommendations for the authors):*

It would have been good for the authors to choose another autoimmune process where the expectation of multiple autoantibodies in the 5-30% of patients are known to occur so that they could actually show the method has the ability to find the common autoantibodies. And if it largely cannot, to define the places where it is best applied. The APS1 is a great example, but this does not expand that finding much, and the other examples don't really provide enough new.

---

## [Author Response]

Essential revisions:1. While all the reviewers recognized that the methodological improvements described have value, there was consensus that they alone do not meet the bar of "substantial developments" or "significant improvements" that a Research Advance require. To move forward there would need to be some application of the new method that reveals important information about an autoimmune disease or an immunological mechanism.

We thank the reviewers and reviewing editor for their suggestions. We have responded to each of the points raised by the reviewers in a point-by-point rebuttal below. Given your concerns regarding the importance of our findings, in this revision we include new data in a new independent cohort which confirms the presence of anti-BTNL8 antibodies, a newly found autoantigen in IPEX through an independent radioligand binding assay. These new results offer novel insight into the antigen-specificity of an autoimmune phenotype in IPEX that has remained poorly understood.

We address the additional concerns below, in response to the public evaluation summary, highlighting and enumerating the many important novel findings and advances made in this work.

Reviewer #1 (Recommendations for the authors):This manuscript may have had two goals – one being to develop methodology for more accurate comparison of autoantibody data across many cohorts with large numbers of controls and the other to gain some information on biological/biomedical relevance.

Thank you, we agree with this – the combined goal of our manuscript is to provide a well-characterized PhIP-Seq protocol that enables screening of many human cohorts for providing new findings in biomedical research. We feel that our manuscript demonstrates all 3 of these items, namely (1) a well-validated protocol, (2) application and validation of the approach across many human disease cohorts, and (3) biological relevance through novel, validated autoantigens PDYN, BEST4, BTNL8, and CGNL1, discovered in APS1, IPEX, IPEX/RAG-deficiency, and MIS-C/Kawasaki disease, respectively.

The authors should more lucidly explain how their alterations of the PhIP-seq approach are more than incremental.

As stated in our introduction, questions of disease-specificity, rare antigens, and antigen overlap can only be addressed in larger, scaled experiments (lines 207-209), which is one of the primary alterations of our protocol. Briefly, most studies to-date have been limited in size, including our own previous work – which, as we demonstrate in this manuscript, both limits the detection of rare antigens and also results in false-positive antigens (lacking disease specificity). The protocol alterations presented here also enable a ~10x larger cohort to be run (in a single experiment), relative to our previous publication, while also ensuring additional advantages including low risk for contamination, benchtop accessibility without requirement for robotics, and low plate-to-plate variability. All of these advances were designed and tested extensively to ensure the highest quality protocol at each step. We have expanded our introduction (lines 210-213) to include an additional instance of description of the advantages and differences in our technique.

One of the biological findings of possible note was the shared hits between KD and MIS-C. Perhaps it offers one way in which the disease relevance of this manuscript could be strengthened. This part could be expanded on a bit with more clinical information examining whether the MIS-C patients with antibodies to CGNL1 had more obvious cardiac findings compared to other MIS-C patients and perhaps consider tapping into other MIS-C cohorts for whom clinical information is also available.

We thank the reviewer for the thoughtful suggestion. Briefly, for CGNL1, the numbers of positive samples were too low to perform statistical testing. Regarding evaluation of cardiac findings in this these samples, we assume this suggestion is referring to coronary artery aneurysm rather than cardiac dysfunction as CGNL1 is an endothelial antigen and not a myocardial antigen. Qualitatively, the positive samples did not have markedly increased cardiac aneurysm z-scores, with sample number limiting statistical analysis. We agree that future studies looking at new cohorts of MIS-C and KD, particularly those enriched for phenotypes of interest including high cardiac aneurysm z-scores (in KD cohorts) and cardiac dysfunction (in MIS-C cohorts), will be of interest. However, this manuscript (1) is focused on technological advances and (2) already screens and validates antigens with significant clinical relevance in APS1, IPEX, RAG1/2 deficiency, and severe COVID-19. We believe the suggestion to apply the technology presented in this paper to a larger cohort of MIS-C highlights the importance of the advances presented in this paper. Indeed, using PhIP-Seq in significantly larger cohorts of MIS-C cases and controls represents one potential important future application of scaled PhIP-Seq.

Reviewer #3 (Recommendations for the authors):It would have been good for the authors to choose another autoimmune process where the expectation of multiple autoantibodies in the 5-30% of patients are known to occur so that they could actually show the method has the ability to find the common autoantibodies. And if it largely cannot, to define the places where it is best applied. The APS1 is a great example, but this does not expand that finding much, and the other examples don't really provide enough new.

We were able to find novel antigens which we validated in 15% and 40% of IPEX patients, respectively. Why this finding does not have novelty or substantial value in the absence of known antigens that occur at the requested frequency of 5-30% is unclear to us. This is especially true given that we have demonstrated that in APS1, many of the novel and interesting antigens can be present in a lower percentage of patients even when orthogonally validated.

In regard to not providing sufficient novel findings, we respectfully disagree, and have summarized the large number of novel findings above.